# Custom-Made Metaphyseal Sleeves in “Beyond” AORI III Defects for Revision Knee Arthroplasty—Proof of Concept and Short-Term Results of a New Technique

**DOI:** 10.3390/jpm13071043

**Published:** 2023-06-25

**Authors:** Yannik Hanusrichter, Carsten Gebert, Marcel Dudda, Jendrik Hardes, Arne Streitbuerger, Sven Frieler, Lee M. Jeys, Martin Wessling

**Affiliations:** 1Department of Tumour Orthopaedics and Revision Arthroplasty, Orthopaedic Hospital Volmarstein, 58300 Wetter, Germany; 2Center for Musculoskeletal Surgery, University Hospital of Essen, 45147 Essen, Germany; 3Department of Orthopedics and Tumor Orthopedics, Muenster University Hospital, 48149 Muenster, Germany; 4Department of Trauma Surgery, University Hospital Essen, 45147 Essen, Germany; 5Department of Orthopedics and Trauma Surgery, BG-Klinikum Duisburg, University Duisburg-Essen, 47249 Duisburg, Germany; 6Department of Orthopedic Oncology, University Hospital Essen, 45147 Essen, Germany; 7Oncology Department, The Royal Orthopaedic Hospital, Birmingham B31 2AP, UK; 8Faculty of Health Sciences, Aston University, Birmingham B4 7ET, UK

**Keywords:** revision knee arthroplasty, AORI, proximal tibial replacement, cone, sleeve, periprosthetic joint infection, tibial bone defect, custom-made

## Abstract

Background: While off-the-shelf cones and sleeves yield good results in AORI type 2 and 3 defects in revision knee surgery, massive longitudinal defects may require a proximal tibia replacement. To achieve the best anatomical as well as biomechanical reconstruction and preserve the tibial tuberosity, we developed custom-made metaphyseal sleeves (CMSs) to reconstruct massive defects with a hinge knee replacement. Methods: Between 2019 and 2022, 10 patients were treated in a single-center study. The indication for revision was aseptic loosening in five cases and periprosthetic joint infection in five cases. The mean number of previous revisions after the index operations was 7 (SD: 2; 4–12). A postoperative analysis was conducted to evaluate the functional outcome as well as the osteointegrative potential. Results: Implantation of the CMS in rTKA was carried out in all cases, with a mean operation time of 155 ± 48 (108–256) min. During the follow-up of 23 ± 7 (7–31) months, no CMS was revised and revisions due to other causes were conducted in five cases. Early radiographic evidence of osseointegration was recorded using a validated method. The postoperative OKS showed a significant increase (*p* < 0.001), with a mean score of 24 (SD: 4; range: 14–31). Conclusion: Custom-made metaphyseal sleeves show acceptable results in extreme cases. As custom-made components become more and more common, this treatment algorithm presents a viable alternative in complex rTKA.

## 1. Introduction

Massive longitudinal bone defects reaching the tibial tuberosity are complex challenges in revision total knee arthroplasty (rTKA), and proximal tibial (PT) replacements may be required to reconstruct the bone defects. However, the literature shows that these procedures have high complication rates and limited function for tumor indication and perform even worse when used for rTKA [1,2,3]. While the AORI classification provides a functional tool to assess defects pre- and intraoperatively and is appropriate to estimate bone loss in the majority of revision knee operations, large meta-/diaphyseal defects can be characterized as “beyond” AORI III. Their treatment remains controversial, and the best treatment rationale is yet to be established [4]. In the three zones of fixation model published by Haddad et al., defects larger than AORI I require solid fixation in zones 2 and 3 [5]. In recent years, metaphyseal augmentation acquired by the usage of cones and sleeves has become “state of the art” [6,7]. Looking at the literature, cones are most useful for defect filling and implant fixations, while sleeves focus more on implant stability and central zone 2 fixation [8]. However, both reconstruction options seem to be superior in terms of implant survival compared to isolated cemented fixation, and the decision depends on the surgeon’s preference in most defects [6,9]. While the usage in “standard” defects might be increasing, complex tibial defects, especially the combination of nearly complete circumferential proximal defects with extensive meta-/diaphyseal bone loss (>3 cm) and erosion of the tibial tuberosity (TT), remain challenging. These defects are only insufficiently described by the AORI classification as ligaments, and the TT and joint stability are not considered. Several factors need to be considered: (I) Standard augmentation often only addresses proximal bone loss and might lead to either an inadequate joint line reconstruction if placed distally or insufficient bone contact if placed in the normal position.

(II) Due to cortical bone defects, sufficient press fit might be unachievable with a cone or a sleeve, as insufficient cancellous bone is present (Figure 1). (III) It is often necessary to compromise between the offset still achievable for “anatomical” reconstruction and the maximum stem diameter for the best possible fixation; consequently, an uncemented tibial stem is often no longer possible due to the insufficient press fit (Figure 1).

Unlike femoral modular megaprosthesis reconstruction [10,11], complete proximal tibial replacement should be avoided, as the necessity of an extensor mechanism excision requires secondary reconstruction, often in combination with a gastrocnemius flap, and is often associated with poor functional outcomes and a high rate of secondary amputation, as published in tumor indications [12]. Treatment options, especially options with the preservation of the TT, in these complex defect situations are sparse. Often, these cases are insufficiently addressed by off-the-shelf metaphyseal augmentation options. Considering the philosophy of custom-made implants in acetabular reconstruction and adapting this to the specialties in rTKA, we developed custom-made metaphyseal sleeves (CMSs) in rTKA to obtain anatomical reconstruction while preserving and augmenting the native extensor mechanism. In a single-stage study, the clinical and radiological outcomes of a newly established therapy algorithm using these implants were evaluated.

As a proof of concept, the results and mistakes were continuously re-evaluated and the design was adapted.

## 2. Therapy Algorithm and Implant Design

Patients presenting with extensive tibial bone loss in all three zones (AORI III) were routinely discussed by experienced surgeons. If the preoperative X-rays showed a mismatch between the off-the-shelf reconstruction methods (cones as well as sleeves), thin-layer CT (<1 mm) scans with artefact suppression were conducted. From these data, 3D models of the knee were compiled, further assessing the defect configuration. This was followed by a 3D assembly of modular off-the-shelf reconstruction methods with the best possible placement to evaluate the treatment options. If the results, with close cooperation between the head surgeon and engineer, were deemed insufficient, custom-made metaphyseal sleeves were designed to allow the best possible fixation while preserving the most bone stock to achieve the maximum highly porous contact area, with a special emphasis on the tuberosity, preserving the extensor mechanism. In addition, joint line reconstruction, oriented on the patellar position, was one of the key factors for the design philosophy. The CMSs were combined with the smallest available modular standard tibial platform of a hinged knee system (MUTARS System, Implantcast, Buxtehude, Germany; Megasystem-C, Waldemar Link, Hamburg, Germany) using cementless off-the-shelf stem fixation. The approved design was manufactured using additive layer manufacturing with a highly porous titanium metal surface to achieve bone ingrowth and biological fixation. Additionally, −4 mm and size-to-size rasps were manufactured to optimize intramedullary defect preparation while also serving as a trial implant (Figure 2). If an extensor mechanism deficiency was suspected or present in the explantation, additional implant augmentation was available through pre-drilled suture holes, similar to the augmentation used for PT.

For each case, individual instructions were assembled, showing detailed images of the necessary bone preparation. Implantation was carried out as a single-stage (*n* = 5) or two-stage (*n* = 5) surgery based on the primary indication. After meticulous tibial debridement, defect re-evaluation was carried out. If the defect was comparable to the preoperative planned design, the implant as well as custom-made rasps were opened. Then, stem as well as metaphyseal sleeve preparation was performed, followed by the cementless implantation of the definitive implant. Small cavitary defects and gaps in the bone/implant interface due to undercutting were addressed using allografts or a ceramic bone substitute. Tibial plateau contact was augmented using PMMA. As increased metal-on-metal wear was present in two of the first patients, detected by serum concentrations (chrome, cobalt, and nickel), an additional PMMA augmentation between the tibial platform and the CMS was created ex situ after the first cases and was fixed with two screws. Postoperatively, X-rays were conducted in all patients. CT scans were only performed for special clinical questions. Load bearing was prohibited only due to secondary parameters (e.g., femoral fixation and extensor mechanism repair), and the CMS allowed direct full load bearing postoperatively. Perioperative antibiotic prophylaxis was administered according to the standardized protocol, with a double-shot administration in the case of prolonged operation time (<120 min).

## 3. Materials and Methods

Between 2019 and 2023, 13 consecutive patients (9 females and 4 males) were treated with a custom-made metaphyseal sleeve in combination with rTKA (*n* = 9 Implantcast GmbH, Buxtehude, Germany; *n* = 1 Waldemar Link GmbH & Co., KG, Hamburg, Germany). The inclusion criterion was a follow-up >6 months (*n* = 10). To evaluate the technique, exclusion criteria were explicitly not defined. A preoperative analysis showed an AORI III defect (meta-/diaphyseal > 6 cm) in all cases that could not ideally be assessed using off-the-shelf cortical cones or sleeves. Therefore, the described implant algorithm was conducted. The indication for revision was aseptic loosening as well as PJI in five cases. The mean number of previous revisions after the index operation was 7 (SD: 2; 4–12). The mean follow-up was 23 ± 7 (7–31) months.

The CMSs were designed with a mean offset of 9.5 (SD: 4; range: 3–15) mm and a length of 85 (SD: 27; range: 62–142) mm. The mean stem length was 200 (SD: 26; range: 160–250) mm. The highly porous surface area was 74.19 (SD: 30.54; range: 55.9–146.4) cm^2^.

All patients agreed to the described method, giving informed consent and permission to submit their data for analysis. Ethical approval was obtained prior to the investigation from the local ethics committee (reference number: 21-10438-KOBO) for nine patients (MUTARS System). For one patient (Megasystem-C), singular individual consent was obtained.

All patients were routinely assessed in outpatient presentations 6 weeks and 3/6/12 months postoperatively, followed by yearly presentations according to the standard treatment protocol. To determine the outcome, the Oxford Knee Score, the range of motion, and the EQ5D were surveyed in all patients.

During the follow-up, revisions and complications (nerve or vessel injury, hematoma, seroma, wound healing problems, and failed primary osteointegration) were recorded.

The early radiographic classification of osseointegration was assessed using radiographic zones, as described in the Knee Society Radiographic Evaluation and Scoring System (KSRESS) [13]. The implant–bone interface in the metaphyseal zone was classified directly postoperatively (complete contact, partial contact, or no contact) as well as during the most recent follow-up (complete osseous integration, partial integration, a stable radiolucent line, or a progressive radiolucent line). Additionally, defect size was evaluated in the CT scan to evaluate the dimensions in which a CMS was needed. To achieve a more detailed assessment of this newly developed implant technique, the press-fit zone between the CMS and the tibial bone was measured using a digital subtraction analysis based on the CT scans in all MUTARS System knees. This was compared to the largest possible off-the-shelf sleeve and cone in a hypothetical, not achievable full-contact situation.

Data analysis was performed using the Statistical Package for Social Sciences Software (IBM SPSS Statistics Version 24, Chicago, IL, USA). Descriptive statistical results were recorded to describe comorbidities, complications, and previous procedures as well as the fixation results. The Shapiro–Wilk test was performed to identify a non-normal/normal distribution. If not mentioned otherwise, the results are stated as means (standard deviation; range). A *t*-test was used for parametric values, and the Mann–Whitney U test was used for non-parametric values. Interobserver agreement was assessed using weighted kappa (linear weighting), and the correlation of the bone contact grades between the postoperative and latest radiographs was analyzed using Kendall’s tau beta. The significance level was set at *p* < 0.05.

## 4. Results

Implantation of the custom-made sleeve in rTKA was carried out in all cases with a mean operation time of 155 ± 48 (108–256) minutes. The mean inpatient stay was 27 ± 15 (12–59) days. The manufacturing time of the CMSs (from first planning until implantation) was 60 (SD: 14; range: 36–76) days.

Femoral implant retention was possible in four aseptic patients. Due to femoral bone loss, modular rTKA with augments was used in one case, a distal femoral replacement was used in four cases, and a total femoral replacement was used in one case. A plastic gastrocnemius flap was necessary in one patient due to insufficient soft tissue coverage. No patient underwent amputation during the follow-up.

In one case, a perioperative tibial fissure distal to the CMS occurred, which was assessed intraoperatively, and internal fixation was performed with cerclage fixation. Partial extensor mechanism deficiency was present in two cases preoperatively and was treated using suture cerclages.

During the follow-up, implant survival was present in all cases; however, all-cause revisions occurred in five patients. In one case, extensive seroma was present postoperatively and was treated with an exchange of mobile parts. Another patient was diagnosed with a capsule insufficiency after extensor mechanism reconstruction, which made an exchange of the mobile parts necessary, followed by a free anterolateral thigh flap. In one case, the initial tibial component was too large, resulting in a capsule insufficiency in combination with acute PJI. This was treated in a second operation using a DAIR procedure in combination with a lateral gastrocnemius flap. In two patients, isolated tibial platform exchange was necessary and was conducted as described in Section 5.

The postoperative Oxford Knee Score showed a significant increase from 9 (SD: 4; range: 2–14) preoperatively to 23 (SD: 4; range: 14–31) postoperatively, with a delta of 14 (SD: 4; range: 8–19) (*p* < 0.001; Mann–Whitney U test). Postoperative active extensor insufficiency was present in two cases, one with 5° and one, after a reimplantation of a total femur prosthesis, with 30°. Case-by-case results are shown in Table 1, and exemplary cases are shown in Figure 3 and Figure 4.

Walking aids were needed in nine patients (*n* = 6 walker, *n* = 2 crutches, and *n* = 1 walking cane). The mean Quality of Life index (EQ5D) was 0.157 (SD: 0.38; range: −0.378–0.75) preoperatively and 0.455 (SD: 0.23; range: 0.272–0.913) postoperatively, with a delta of 0.297 (SD: 0.38; range: −0.3–1.1).

The postoperative metaphyseal bone contact and final follow-up osseointegration according to the KSRESS are shown in Table 2. The interobserver reliability was substantial (weighted kappa: 0.774, *p* < 0.001), and a comparison between the immediately postoperative radiograph and the osseointegration at the latest follow-up showed a significant correlation (Kendall’s tau-b: 0.591, *p* < 0.001).

A three-dimensional defect analysis showed a mean volume of 75.46 (SD: 36.49, range: 38.48–162.98) cm^3^. A digital subtraction analysis resulted in (I) a mean contact area (highly porous structure to bone) of 57 (SD: 20.42; range: 39.97–94.10) cm^2^ (Figure 5) and (II) a relative (bony contact area/highly porous surface area) achieved contact area of 78 (SD: 12; range: 61–92) %. A comparison to the maximum possible sleeve in an ideal, not achievable “full-contact” situation resulted in (I) a delta of 39.31 (SD: 19.25; range: 21.73–74.14) and (II) a relative increase of 297 (SD: 71; range: 208–446) % compared to the sleeve surface (Figure 5).

## 5. Design Adaptations

The preoperative planning of a CMS is a complex and challenging task, even compared to custom-made acetabular (rTHA) or custom-made cone (rTKA) components, as several factors need to be evaluated cautiously [14]. Due to the monobloc construct, implant rotation as well as offset and alignment have to be planned precisely preoperatively, inferring a small margin of intraoperative orientation. Moreover, as a press fit should be achieved around the CMS and the stem, the planned components need to be balanced between “best-fit” and tolerable bone stress levels to prevent intraoperative fractures. These factors lead to a demanding preoperative planning process and require close cooperation between the engineer and the surgeons. Moreover, profound experiences in rTKA as well as modular implant design are vital. We want to mention several key mistakes during the study process that have led to subsequent design changes to prohibit these in other centers.

(I) In two cases where isolated knee CT scans were used, postoperative malrotation was present. This led to elevated metal-on-metal wear with increased systemic chrome and cobalt levels, as measured by a serum analysis. Consequently, the tibial platform was exchanged with a custom-made rotated platform that was augmented with PMMA in the CMS–platform interface after the interoperative assessment showed metal wear on both interfaces (Figure 5). Subsequently, the systemic metal levels normalized and the functional outcome increased significantly. Therefore, we advocate conducting preoperative thin-layer CT scans with rotational imaging of the hip and ankle joints (Figure 5). To prohibit the increased wear, even in correctly rotated plateaus, PMMA augmentation was carried out in all subsequent cases. Additionally multi-point fixation should be considered for the tibial platform and has been conducted with four screws in the new CMS, which was not included due to a follow-up <6 months.

(II) In one male patient, a standard tibial platform was chosen, unlike the small platform used for the other patients, resulting in a capsule insufficiency, as soft tissue shortening was presented due to multiple preoperative revisions. We advise using the smallest possible components, as soft tissue handling is a key factor that is not featured in the digital preoperative planning process.

## 6. Discussion

The osteointegrative preservation and augmentation of a stable tibial tuberosity is one of the key factors in revision arthroplasty to achieve a satisfactory postoperative result. In a meta-review, Zanirato et al. included 37 studies publishing long-term survival rates of 97.3% for cones and 97.8% for sleeves, which are well-recognized treatment options for AORI type 2 and 3 defects, especially compared to isolated cemented implantation [5,6]. Adding to this, Mancuso et al. showed excellent outcomes when reviewing a multitude of studies using metaphyseal components for AORI IIB and III defects [15]. Looking at the recent literature, 3D printed, highly porous titanium components show superior results, with improved osteointegration, while simultaneously reducing the risk of infection [16,17]. Furthermore, as England et al. published, the structure shows a promising osteointegration potential, even in a short follow-up [7]. The custom-made sleeves are manufactured using the same structure as in highly porous sleeves and partial pelvic replacements; therefore, the promising results in these implants are expected to be transferable [17,18].

The key factors for successful reconstruction in large tibial bone defects “beyond AORI III” need to be emphasized: (I) the osteointegrative preservation of the extensor apparatus, with special consideration of the TT; (II) the biomechanically appropriate reconstruction of the joint line; and (III) the maximum possible osteointegration for long-term implant survival. Additionally, due to the custom tibial offset, anatomical tibial plateau positioning as well as native mechanical axis reconstruction can be achieved. Thus, in these cases, custom-made implants should be considered, as these offer a viable solution to achieve satisfying outcomes in these patients with intact TTs.

Looking at the recent literature, custom-made cones have been published as a treatment strategy. Burastero et al. published the usage of 8 tibial cones in 11 patients with a mean follow-up of 26 months. They showed a low rate of complications (*n* = 1) and only reported functional outcomes in three patients (OKS: 41;43;38) [14]. Using a similar treatment approach, Savov et al. reported results in 10 patients with 9 custom-made tibial cones, showing a low complication rate (*n* = 2) at a mean follow-up of 21 months, though no functional outcome parameters were reported [19]. Li et al. published a study using a similar method in seven patients with good functional outcomes at a follow-up of 25 months (postoperative HSS: 78 (70–83) (*p* < 0.01)) [20].

While the functional outcomes of our patient population seem to be slightly reduced compared to the literature (postoperative mean OKS: 24), the defect size does seem significantly larger. However, as no surface area or defect volume analyses were reported in any of the mentioned studies, defect evaluation remains limited. Additionally, due to the usage of cones, stem size and offset correction are limited.

Life quality indexes were not present in any of the described literature; hence, we can only state our promising results, with a mean EQ5D increase of 0.297, similar to published studies after “standard” rTKA [21].

While the literature suggests that cones and sleeves might be used interchangeably in smaller defects, we have specifically opted for the use of custom-made sleeves and not cones in our collective due to several factors. The implantation of a CMS should only be indicated when sufficient epiphyseal fixation is no longer possible and metaphyseal fixations are highly compromised [8,19]. In these cases, long-term implant survival can only be achieved by osteointegrative fixation in zones II and III [5]. Furthermore, in these complex cases, a high offset is often required to achieve anatomical and biomechanical reconstruction, as seen in our collective, with a mean offset of 9.5 (SD: 4; range: 4–15) mm. Using a custom-made cone achieving diaphyseal reconstruction (zone III) requires a cemented implant due to the limited stem size, as seen in the collective by Savov et al. [19]. As osteointegrative fixation in zone II is mandatory for the long-term survival of the prosthesis, an exacting cementing technique must be consistently conducted.

Additionally, the defect analysis resulting in a mean CMS length of 85 mm shows that the proximal defect is almost always accompanied by expansive meta-/diaphyseal bone loss, making the cementless contact between custom cones and bone even more difficult.

Considering these factors, we advocate for the usage of custom-made sleeves to obtain the “best of both worlds”: maximized implant stability with optimal press-fit defect filling [6]. This results in a cementless reconstruction with the best possible fixation in zones II and III and less limited stem size while eliminating the risk factor of a cemented interface on the highly porous structure.

The radiological evidence of osseointegration was encouraging (Table 1). In a more detailed analysis, only two zonal aspects decreased during the follow-up; all other measurements remained stable or increased slightly. As mentioned, TT preservation is essential, as seen in Table 1, with full osseointegration in 7 out of 10 patients in zone 3 A. The results seem comparable to the off-the-shelf implant osseointegration potential reported by England et al. [7]. To our best knowledge, a three-dimensional tibial defect analysis has not yet been published. We can only state our results, with a mean defect of 75 cm^3^. As at least a doubling could be achieved in all patients compared to off-the-shelf implants, custom-made implants are justified in these cases.

Extensor mechanism insufficiency should be cautiously examined preoperatively, as a CMS should only be implanted if a viable extensor mechanism with bony TT preservation is present. However, a relative deficiency can be addressed using fiber-wire augmentation for smaller structural defects (*n* = 2).

Unlike custom-made implants, the usage of off-the-shelf stacked tibial cones may be an alternative treatment strategy in some cases, but this requires a high level of expertise and excellent cementation technique and provides, as with custom cones, only limited offset correction. However, the functional outcome of this technique remains unclear, as unlike its utilization in femoral bone loss [22], only case reports have been published [23].

Since PT is the alternative treatment option in this collective, a comparison of the algorithms should be discussed. However, results for proximal tibia in rTKA are sparse. The studies by Hoell et al. and Fram et al. show limited functional outcomes and extensor lag in 5 out of 6 patients each, ranging from 5 to 45 degrees [2,3]. The main indication for a proximal tibia is oncological orthopedic surgery, and several publications have reported outcomes after PT implementation. However, even in these cases, extensor mechanism insufficiency is common. Biau et al. published a rate of 11% in 91 patients [24]. Additionally, implant survival is alerting, even with an oncological indication. While Hardes et al. published encouraging results for different oncological indications in a single-center analysis [12,25], Sacchetti et al. recently showed a survivorship of 59% in a meta-analysis, with a mean extensor lag between 7 and 30 [1].

As a CMS is only needed in multiply revised and compromised cases, the expected postoperative results should be discussed in detail preoperatively, as a “restitutio ad integrum” is often not feasible and secondary parameters often need to be taken into account.

Accordingly, salvage therapy should be considered in each case. Arthrodesis always remains as an option; however, the functional outcome should be discussed thoroughly [26].

The manufacturing of a custom-made implant takes six to eight weeks with an experienced team. This should be noted to the patient. Therefore, urgent operations, e.g., imminent fractures, are not possible with a definitive treatment. In aseptic conditions, a two-stage procedure might be discussed. CT scans with a PMMA spacer provide better bone visualization than artefact-suppressed CT scans with a TKA. However, secondary interval factors (immobilization and the risk of thromboembolism) should be taken into account.

While no CMS was revised, we report prima facie a relatively high all-cause revision in 5 out of 9 cases (55%). Considering the patient collective and the number of previous revisions (mean: 7), this is concordant with the recent literature [27]. In consensus with Rajgor et al., we also report infection control in one case after staged DAIR [28].

Several limitations need to be mentioned. As this analysis was conducted as a single-center study, the indications as well as the fixation methods were heavily influenced by the department’s philosophy. In addition, the number of patients was sparse due to the rare indication. However, as this treatment presents a new algorithm, the follow-up of 23 months, as well as the number of patients, seems sufficient, especially considering the alternative treatment strategy of PT in rTKA, where only small case series have been published. Each CMS was an exceptional case. Correspondingly, the defect structure showed a high level of heterogeneity, and a detailed classification of these defects beyond AORI III is not yet established due to the low number of cases in a single department. Nevertheless, as re-revisions seem to be increasing, it is expected that surgeons will be confronted more and more with these large defects in the future. Therefore, a multi-center study evaluating large tibial defects seems to be necessary to further establish standardized treatment algorithms and diagnostic approaches. Additionally, custom-made implants, especially for rTHA, are becoming more common; thus, transmission to rTKA is expected and will enable larger studies. Besides the surgical and patient parameters, costs must be considered. Regardless of the manufacturer, 4–5 times the costs of an “off-the shelf” metaphyseal augment (either cone or sleeve) can be expected.

Additionally, besides the complex tibial reconstruction, femoral bone loss is presented in nearly every case due to the multitude of surgeries. Correspondingly, there were only two condyle-preserving femoral rTKA components in our collective. This should be considered, especially for the long-term follow-up. While Wyles showed an acceptable functional outcome with a mean KSS of 71 in cases with isolated distal femoral replacement, the revision rate remains high [29].

## 7. Conclusions

Exceptional metaphyseal bone loss (zone II) is often inadequately addressed using standard implants. CMSs offer a viable solution to achieve sufficient press fit and osteointegration.CMSs allow adaptation to the existing defect with minimal possible bone loss and bony preservation of the tibial tuberosity.The cementless design allows for optimal press fit in all three zones while simultaneously restoring the mechanical axis and joint line.

## Figures and Tables

**Figure 1 jpm-13-01043-f001:**
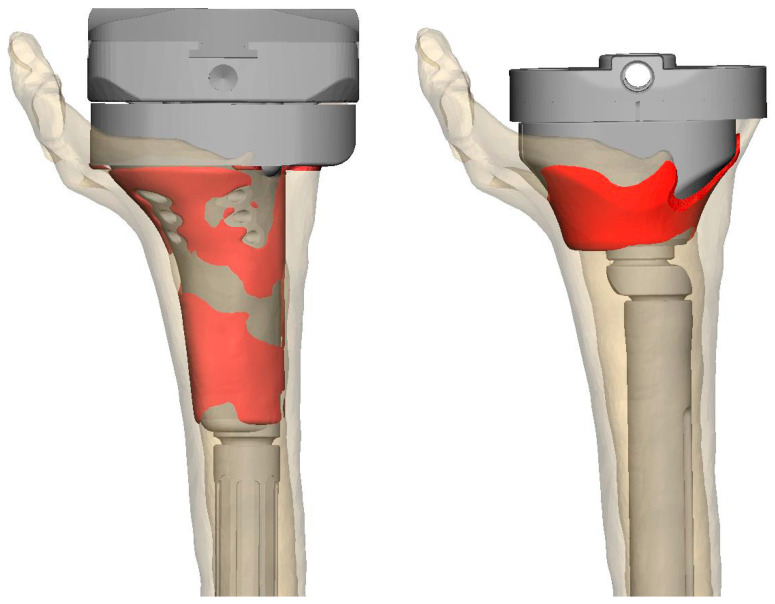
Exemplary surface analysis in a patient not included due to follow-up <6 months. In total, 41.70 cm^2^ of 56.47 cm^2^ (73%) in the CMS achieved osseus contact, compared to 19.48 cm^2^ of 36.24 cm^2^ (53%) in an optimally positioned cortical cone. The patient was allowed full weight bearing postoperatively and achieved a flexion of 0–10–75° 2 months postoperatively after simultaneous quadriceps augmentation.

**Figure 2 jpm-13-01043-f002:**
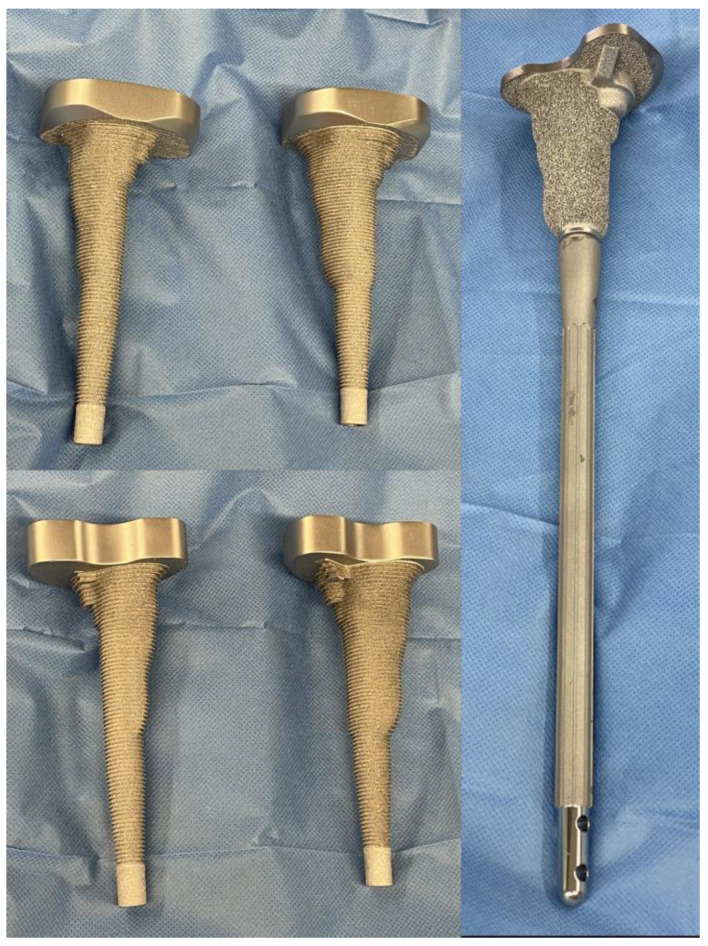
Custom-made rasps (−5 mm and size-to-size) as well as the definitive implant.

**Figure 3 jpm-13-01043-f003:**
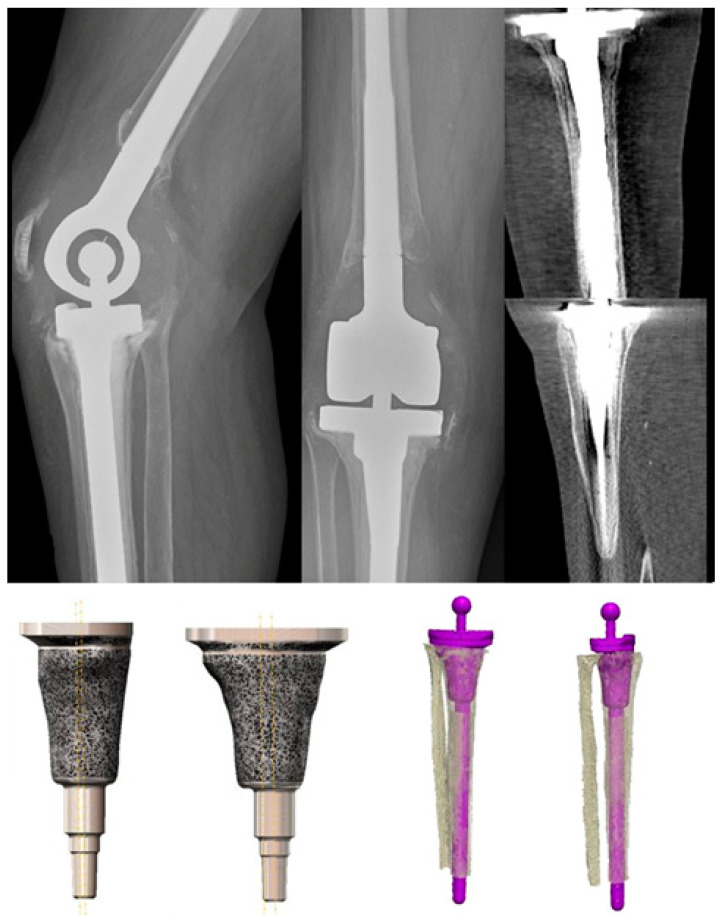
Aseptic loosening in a 77-year-old female patient (case 2). The CT scans show the combined meta-/diaphyseal AORI III defect, the planned individual component, and the final postoperative implant positioning (purple implant). The yellow lines represent the planned off-set.

**Figure 4 jpm-13-01043-f004:**
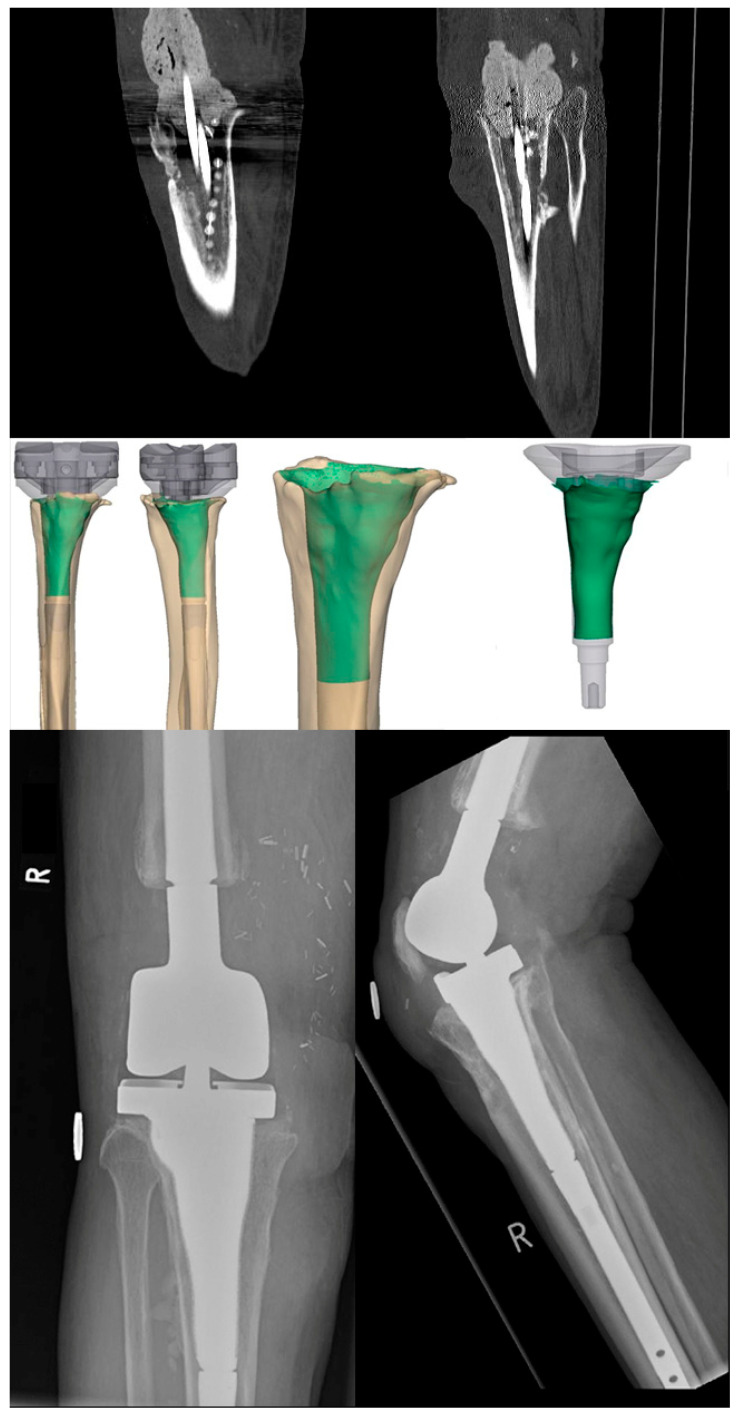
Case number 9, an 80-year-old male patient with a multi-pathogen PJI treated in a two-stage exchange, showing the extensive defect, the contact analysis, and the postoperative results.

**Figure 5 jpm-13-01043-f005:**
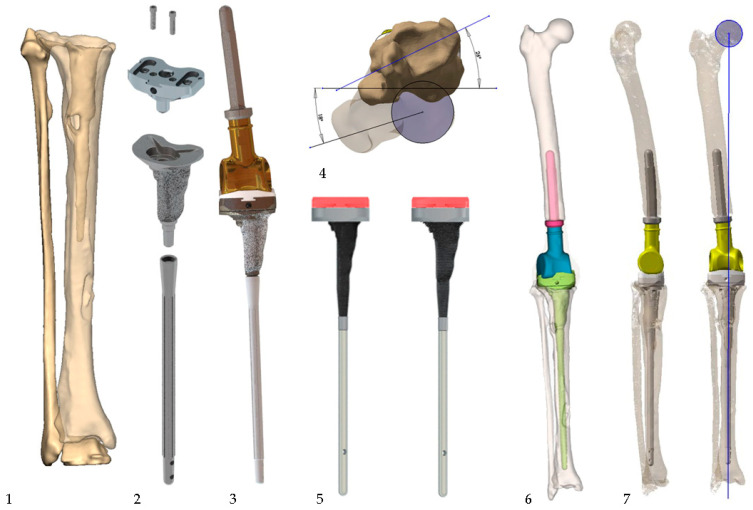
Case number 10, a 71-year-old male patient with a Staphylococcus epidermidis PJI treated in a two-stage exchange. Image 1: perioperative defect. Image 2: tibial assembly with a small modular tibial platform fixed with 2 screws. The interface between the CMS and the tibial platform was augmented with PMMA, and additional screw fixation (e.g., 4 instead of 2) was considered. Image 3: final prosthesis assembly. Image 4: preoperative rotational measurement in a 3D model. Image 5: −5 mm and size-to-size rasps with a trial assembly. Image 6: digital planning of the prosthesis. The rotation was not yet corrected. Image 7: AP and lateral views with axis measurement. Extensor mechanism pilot holes are shown in the proximal CMS.

**Table 1 jpm-13-01043-t001:** Patient overview.

Patient	Indication	Tibial Component Preoperatively	Femoral Component Postoperatively	ROM Postoperatively	OKS Pre-/Postoperatively	Pain (NRS) Pre-/Postoperatively	CMS Contact Area (% of Total Surface Area)
1	PJI	Hybrid fixation Cone	TFR	0-30-80	2/20	9/2	39.97 cm^2^ (71%)
2 *	Aseptic loosening	Cemented Augmented	DFR	0-0-90	12/20	7/2	52.74 cm^2^ (92%)
3 *	Aseptic loosening	Cemented Augmented	DFR	0-0-90	14/24	8/2	40.94 cm^2^ (72%)
4 *	Aseptic loosening	Cemented Augmented	rTKA	0-0-110	7/22	4/1	55.27 cm^2^ (91%)
5	Aseptic loosening	Hybrid fixation Augmented	DFR	0-0-75	9/27	6/2	94.1 cm^2^ (64%)
6	PJI	Hybrid fixation Augmented	rTKA	0-0-110	12/31	5/1	53.01 cm^2^ (0.88)
7	Aseptic loosening	Cemented Augmented	DFR	0-5-90	9/20	6/3	89.05 cm^2^ (90%)
8 *	Aseptic loosening	Cemented Augmented	DFR	0-0-90	10/25	4/1	
9	PJI	Uncemented Sleeve	DFR	0-0-50	1/14	8/2	46.3 cm^2^ (0.61%)
10	PJI	Hybrid Fixation	DFR	0-0-90	17/28	7/2	41.7 cm^2^ (73%)

ROM—Range of motion; OKS—Oxford Knee Score; CMS—Custom-made metaphyseal sleeve; PJI—Periprosthetic joint infection; TFR—Total femur replacement; DFR–Distal femur replacement. * isolated tibial exchange.

**Table 2 jpm-13-01043-t002:** KSRESS analysis.

		KSRESS Zone			
		3 M	3 L	3 A	3 P
Immediately Postop.	Complete Contact	2	6	7	4
	Partial Contact	8	4	3	6
	None	0	0	0	0
Latest Follow-up	Full Ingrowth	4	6	7	4
	Partial Ingrowth	6	4	3	6

CMS contact on immediate postoperative X-rays and osseointegration on the latest follow-up X-rays. 3 M—zone 3 medial; 3 L—zone 3 lateral; 3 A—zone 3 anterior; 3P—zone 3 posterior.

## Data Availability

The datasets used and/or analyzed during the current study are available from the corresponding author on reasonable request.

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
