# Peer review of "Custom-Made Metaphyseal Sleeves in “Beyond” AORI III Defects for Revision Knee Arthroplasty—Proof of Concept and Short-Term Results of a New Technique"

_jpm, 2023, doi:10.3390/jpm13071043_

Round 1
Reviewer 1 Report
Very interesting cases for the arthroplasty surgeon. well written
I would add a comment regarding the costs of those implants as it is significant in some of the countries.
As an arthroplasty surgeon, it was a very interesting and well written article to read.
There are many articles discussing this topic - tibial bone management in total knee revision. However I couldn't find an article specifically going into details with custum made implants so I would consider it as an original and unique article.
My recommendation for this article is to accept it with a minor revision as I wrote in my review.
My revised review Is the following:
"
This article describes the challenging treatment of tibial bone loss in knee arthroplasty with custom made implants. Every case is unique and challenging to the arthroplasty surgeon
I would consider adding a brief description of the costs for those specific implants as it is a barrier for many surgeons coming to use those implants.
What is the average time waiting for the implant to be manufactured for the patient?
What was the post op protocol for patient I. Terms of weight bearing? Range of motion?
Prophylactic Antibiotics treatment for the non infected patients?
"
Author Response
Please see the attachement.

Reviewer 2 Report
The article "Custom-made metaphyseal sleeves in “beyond” AORI III defects for revision knee arthroplasty Proof of concept and short-term results of a new technique is a well organized study presenting a new treatment algorithm for complex rTKA.
1. L118-119: "Tibial plateau contact was augmented using PMMA, additionally due to increased metal-on-metal wear between the tibial platform and the CMS" Did the authors make an attempt to quantify the wear by any imaging technique. How was it concluded that the wear was increased.
2. What statistical technique is used should be clearly mentioned. Suggest the authors refer the below article and improve the statistical analysis
"https://journals.physiology.org/doi/pdf/10.1152/ajplung.00238.2017"
3. Please discuss the " normality assumption" made in the current context of data.
4. L252- "This led to elevated metal-on-metal wear with increased systemic Chrome and Cobalt levels". How the increased cobalt and chrome levels assessed? Did the authors try to co-relate the same with the activity level of the patient.
5. As the single centre study has limitations, what are the authors future plans to conduct multi centre study.
5. Suggest the conclusions are written in a crisp and clear manner. Presenting them in bullet points would be better for improved readability.
The general standard of English used in good. A check on sentence construct at some places would improve the quality.
